# How Chemometrics Can Fight Milk Adulteration

**DOI:** 10.3390/foods12010139

**Published:** 2022-12-27

**Authors:** Silvia Grassi, Maria Tarapoulouzi, Alessandro D’Alessandro, Sofia Agriopoulou, Lorenzo Strani, Theodoros Varzakas

**Affiliations:** 1Department of Food, Environmental and Nutritional Sciences (DeFENS), Università degli Studi di Milano, Via Celoria, 2, 20133 Milano, Italy; 2Department of Chemistry, Faculty of Pure and Applied Science, University of Cyprus, P.O. Box 20537, Nicosia CY-1678, Cyprus; 3Department of Chemical and Geological Sciences, University of Modena and Reggio Emilia, Via Campi 103, 41125 Modena, Italy; 4Department of Food Science and Technology, University of the Peloponnese, Antikalamos, 24100 Kalamata, Greece

**Keywords:** fraud, authentication, dairy, clustering, classification, regression, validation

## Abstract

Adulteration and fraud are amongst the wrong practices followed nowadays due to the attitude of some people to gain more money or their tendency to mislead consumers. Obviously, the industry follows stringent controls and methodologies in order to protect consumers as well as the origin of the food products, and investment in these technologies is highly critical. In this context, chemometric techniques proved to be very efficient in detecting and even quantifying the number of substances used as adulterants. The extraction of relevant information from different kinds of data is a crucial feature to achieve this aim. However, these techniques are not always used properly. In fact, training is important along with investment in these technologies in order to cope effectively and not only reduce fraud but also advertise the geographical origin of the various food and drink products. The aim of this paper is to present an overview of the different chemometric techniques (from clustering to classification and regression applied to several analytical data) along with spectroscopy, chromatography, electrochemical sensors, and other on-site detection devices in the battle against milk adulteration. Moreover, the steps which should be followed to develop a chemometric model to face adulteration issues are carefully presented with the required critical discussion.

## 1. Introduction

Milk and milk products provide the human body with valuable nutritional components such as proteins, carbohydrates, vitamins, minerals, οrganic acids, and fat [1,2]. Milk’s high protein content has attracted many consumers, making it a popular nutritional commodity [3]. The increasing consumption of milk and dairy products leads to many cases of adulteration [4,5]. A range of possible milk adulterants is described by Nascimento et al. [4].

The prices of milk differ primarily depending on the type of animal from which they come, whereas its availability is significantly affected by the season. These two factors are enough to cause problems in its market, as practices of replacing it with cheaper milk are common [6]. Goat’s milk shows a nutritional profile superior to that of cows, as a result of which it is a priority for consumers not only in traditional dairy products such as cheese and yogurt, but also in liquid form. Its low production combined with its beneficial nutritional content makes this category of milk an attractive target for adulteration. Goat’s milk is easily mixed with water, whey as well as cow’s milk which is much cheaper. The latest fraud is increasingly worrying people because of their sensitivity to lactose and the allergic disorders that can be caused by cow’s milk proteins [7]. An equally important adulteration is related to the substitution of goat’s milk with sheep’s milk. In this case, the lower price of goat’s milk compared to sheep’s milk pushes the producers to this adulteration [6].

Fraud in milk production is carried out by admixture or substitution of inferior substances and sometimes dangerous products. The economically motivated adulteration (EMA) is the most important, aiming to gain profit by the addition of extraneous water, glucose or other sugars, non-dairy proteins such as soybean and pea protein isolates [8], various substances such as melamine, urea, maltodextrin, cheese whey (a byproduct of cheese production) [9], hypochlorite, dichromate, salicylic acid [10], and reconstituted milk powders to correct protein and/or density values [11]. A famous case of adulteration was recorded in China in 2013 when the substance melamine was detected in milk powder in infant milk products, which was added to increase the apparent protein content, with dramatic consequences for public health [12].

The deliberate addition of formaldehyde to raw milk is also illegal and considered a major adulteration, which aims to increase the shelf life of milk at room temperature. High moisture content is responsible for the rapid spoilage of milk. Therefore, formaldehyde provides preservative and antiseptic properties, and the ability to improve the appearance including the smell of milk. Furthermore, formaldehyde is toxic at low concentrations and is classified as a human carcinogen by the International Agency for Research on Cancer (IARC) [12,13].

Another form of adulteration is the replacement of milk fat with vegetable fats of lower economic value [14]. Among others, soybean oil has been mentioned in the adulteration of milk [15]. In addition, the recent EU regulations for foods designated as PDO (protected designation of origin), PGI (protected geographical indication), and TSG (traditional specialty guaranteed) require the inclusion on the label of the geographical origin of food. In the case of dairy products such as cheeses produced in a defined area with specific physicochemical and sensorial features, their geographical origin is put forward as an important indication [16].

Chemometrics plays a dominant role in the field of food adulteration as it relates a multitude of chemical analytical characteristics to the qualitative and quantitative analysis of food [17]. Deriving a fingerprint of each sample and reflecting its complex chemical composition could be a way to solve such difficult analytical tasks. Then, chemometric techniques can be used to develop classification models to classify samples into authentic/adulterated ones, or regression models aiming at quantifying a specific adulterant [8,18,19,20,21]. In this direction, both specific and non-specific fingerprinting can be implemented. Specific chemical analysis is based on the detection of organic species, mainly achieved by chromatographic techniques.

The non-specific fingerprinting approach relies on the implementation of instrumental methods to obtain a multivariate description of the chemical composition of the sample. These non-specific fingerprints can be obtained by different methodologies such as Fourier transform infrared spectroscopy (FT-IR), mid-infrared spectroscopy (MIR), Raman spectrometry, nuclear magnetic resonance (NMR), or mass spectrometry [22]. All these methodologies have been used in studies, which are relevant to authenticity and chemometrics in milk and dairy products [23,24,25]. In addition, near-infrared (NIR) spectroscopy has been used by several researchers to detect various forms of adulteration in both cow’s milk and cow’s milk products [26,27,28].

Vibrational spectroscopic techniques are rapid, low-cost, and non-destructive tests that require only limited training for processing. Results are evaluated using chemometric models to extract meaningful information that distinguishes different and significant groups by removing redundant data [29].

Data processing can be completed by principal component analysis (PCA) since it is amongst the most fundamental methods for multivariate data exploration [18]. PCA has been used along with other methodologies to help to differentiate fresh milk and reconstituted skim milk powder samples [11].

kNN (k-nearest neighbor), PLS-DA (partial least squares-discriminant analysis), and SIMCA (soft independent modeling of class analogy) are the most popular classification methods [30]. kNN and PLS-DA have been used for the detection of various types of adulteration, such as water, urea, cow’s whey, and cow’s milk in goat’s milk samples [31]. SIMCA could also be employed to model the class of fresh types of milk. When addressing a specific adulterant quantification, the goal could be achieved by means of partial least squares (PLS) regression analysis, as demonstrated for the prediction of fresh milk adulteration with reconstituted skim milk powders [11].

Finally, in order to validate a chemometric approach, a sampling strategy should be followed taking into account the size and the representativeness of the sample along with intrinsic variability [32]. Sampling is closely associated with robustness and reliability. Other key parameters of authenticity and fraud not to be ignored are the heterogeneity of a food matrix and the presence of an undeclared substance to the geographical origin discrimination.

In this framework, the aim of this work is to give an overview of the recent application of different chemometric techniques—from clustering to classification and regression applied—to several analytical data—encompassing spectroscopy, chromatography, and electrochemical sensors—to fight milk adulteration. Further, a critical discussion is presented to schematize the steps which should be followed to develop a chemometric model to face adulteration issues.

## 2. Chemometric Approaches

### 2.1. Clustering

The definition of “cluster analysis” or “clustering” encompasses the techniques which split a set of samples (observations) into several groups or clusters. The outcome is usually represented as a vector of data, or a point (scatter) in a multidimensional space [33]. Clustering falls in the general category of unsupervised pattern recognition and numerical and mathematical taxonomy [33,34]. Natural grouping of data takes place based on some inherent similarity, as clustering is performed without any group labels, and this justifies the unsupervised pattern recognition [33,35]. Furthermore, it takes place based on similarities of the samples within the same group and others in different groups. Therefore, homogeneity is dominant within the same groups [34]. In practice, the most common approach to define similarity is the distance among the patterns; by lowering the distance (e.g., Euclidean distance which is a well-used dissimilarity measure) between the two objects, higher similarity and vice versa will be obtained [35,36].

Clustering is a valuable component of data analysis or machine learning-based applications such as regression, prediction, data mining, etc. [35]. Saxena et al. (2017) [35] stated that there are various ways to categorize clustering methods because it is difficult to define a cluster. In their paper, they suggested division into two different groups such as hierarchical and partitioning techniques, or in three categories based on application, density-based methods, model-based methods, and grid-based methods.

Hierarchical methods initially group the objects into small clusters of some samples, and these are next grouped into larger clusters, thus a dendrogram is produced, which is a tree-based depiction of each observation [36]. Optimization- partitioning methods split the samples into a few groups to optimize a particular feature e.g., total within-group distances. In this category, algorithms like k*-means clustering*, *Fuzzy* c*-means clustering*, etc., are included [33,34,35]. Density-based clustering is focused on the probability that data objects are drawn from a specific probability distribution and the overall distribution of the data is assumed to be a mixture of several distributions. Data points can be derived from different types of density functions (e.g., multivariate Gaussian or *t*-distribution), or from the same families but with different parameters. Model-based clustering works by detecting feature details for each cluster, where each cluster represents a concept or class. Decision trees and neural networks are the two most frequently used methods in this category. Grid-based clustering divides the space into a finite number of cells that make a grid structure on which all the operations for clustering are performed [35].

Recently, many evaluation criteria have been developed, and these are internal and external. Internal quality parameters include the sum of squared error, scatter criteria, Condorcet’s criterion, the C-criterion, category utility metrics, and edge cut metrics. External quality criteria are related to the mutual information-based measure, Rand index, F-measure, Jaccard index, Fowlkes–Mallows index, and confusion matrix [35].

Clustering is applied to perform data reduction or compression for handling huge loads of data. It helps in compressing data information by grouping them into different sets of clusters. This helps us to choose what is useful or not by saving time from data processing along with data reduction [35]. Other uses contain data mining, document retrieval, image segmentation, and pattern classification [33].

In order to explore the use and development of clustering methods recently, Table 1 has been prepared to summarize the studies related to milk adulteration and authenticity.

Regarding milk adulteration studies, Cirak et al. [37] focused on determining milk species adulteration by using FTIR. HCA was conducted based on Ward’s algorithm after having calculated the initial derivate by using a standard method. The produced 2D-dendrogram indicated that the types of origins (sheep, cow, and water buffalo origin, and adulterated samples in binary mixtures) were clustered correctly. Minetto et al. [36] applied HCA to detect urea in raw bovine milk samples, and the Euclidean distance was used to build the dendrogram. HCA helped them to find the more appropriate number of clusters which was used later in the classification of the samples. Vinciguerra et al. [38] used HCA as an exploratory treatment on the pre-processed measurements obtained by FTIR-ATR. By using both the Euclidean distance and Ward’s method, a dendrogram was generated, however no pattern related to the caseinomacropeptide concentration was observed in the dendrogram, and multivariate regression was followed. Qualitatively, the adulterated groups with caseinomacropeptide were separated correctly in 3 groups: raw milk, skimmed milk, and semi-skimmed milk. Adulteration with melamine and urea in cow’s milk was also studied by Ezhilan et al. [39], who developed an electrochemical biosensor to detect the two adulterants simultaneously. HCA application was useful to study the interrelationship of the factors affecting the model for measurements taken by using various combinations of concentrations of the adulterants. Mostafapour et al. (2021) [40] used a colorimetric array device. The authors commented that even if there are differences in the colorimetric schemes of the analytes, it is not a proper manner to group the samples after visual examination, thus chemometrics is used to perform the clustering. The HCA dendrograms showed highly accurate clustering of the studied carbonyl compounds, particularly eight different aldehydes and ketones. In addition, HCA showed that one sample from formaldehyde and one sample from acetophenone has been misclassified. Li et al. [41] used NMR to detect the metabolites as markers of different milk types. Clustering analysis (CA) was very useful as it provided similarities for the same species of milk as well as variations in different milk species by applying the minimum distance method. CA also separated the three milk types and showed that NMR and metabolites can differentiate these milk products. Sowmya et al. [42] during the pre-processing steps applied cluster analysis, i.e., the k-means clustering algorithm. The algorithm proceeded by calculating the centroid point of the dataset and the groups’ mean points to build the new groups required. The aim was to see the grouping of samples, to identify the similarities in the same categories, and to check if the adulterants can be clustered by using raw spectra. Intraclass variation was performed.

Regarding milk authenticity, Souza et al. [43] studied the metal profile of powder and liquid milk samples to differentiate them based on the type of milk. HCA successfully confirmed the initial outcome of PCA, and it allows the visualization of a sample’s trend to form two groups. Whole cow powder milk, whole goat powder milk, skimmed cow powder milk, and milk compounds powder fell in the first group due to their similar composition. A sample from the last group clustered at a longer distance from its group due to the high content of Zn. The second group consisted of whole and skimmed cow liquid milk and some yogurts. Rodriguez-Bermudez et al. [44] by applying HCA revealed a correct clustering based on the type of milk, organic vs. conventional. It was obvious that the variables (metal content) in both the organic and conventional sets were distinct. To determine the geographical origin, Zain et al. [45] measured the metal content of milk samples and due to different environmental conditions, and the samples clustered successfully by HCA. Ca, Na, Fe, Zn, Mn, K, Ba, and Mg are the metals that were significant for the samples’ grouping regarding geographical origin. Xu et al. [46] worked also in terms of geographical origin by measuring isotope ratios, metals, and fatty acids and then by applying HCA. δ^18^O measurements were taken by having the milk in the fluid state, but for δ^13^C, δ^15^N, and elemental and fatty acid measurements lyophilization took place. HCA aided to picture the correlation between the sample and each variable as HCA heatmaps were created. In addition, geographical origin was the target of Amenzou et al. [47], who studied the ^13^C/^12^C, ^15^N/^14^N, ^18^O/^16^O. The application of HCA was very important to visualize the samples in 3 important clusters. The stable isotope ratios analysis in combination with chemometrics showed a very good capability to indicate the geographical origin of milk. In a similar study, Podkolzin and Solovev [48] used HCA and the k-mean clustering algorithm and both methods showed an equal number of clusters with almost the same content. Karrar et al. [49] used HCA to evaluate the similarity in terms of sn-2 and sn-3 fatty acids in different milk-origin samples. HCA heatmaps were produced to present the content of sn-2 and sn-3 fatty acids in the samples. Bhumireddy et al. [50] applied HCA to group the samples based on intrinsic similarities in their GC-MS measurements. HCA heatmaps were produced using the log-transformed and normalized values of the relative abundance of 17 amino acids, and their high and low expressions in each sample were presented with different colors. Tan et al. [51] employed HCA to proceed to the clustering of the different biomarkers (peptides, lipids, and nucleic acids) and to demonstrate the chemical properties of the important metabolites. It must be also noted that the results indicated that the processing that takes place to produce milk powders influences the nutritional loss of peptides and lipids. HCA heatmaps showed that nutritional components were found to be in lower concentrations in reconstituted milk compared to ultra-high-temperature milk. Couvreur and Hurtaud [52] studied the parameters of fat globule characteristics (diameter, membrane surface, and yield), fat, protein, fatty acids, and calcium content in milk concerning diet composition, milking frequency, breed, stage of lactation, parity and residual/cisternal milk. Based on the principal components of PCA, HCA was performed which indicated 4 independent clusters of milk. A minor relationship was observed between fat content and fat globule diameter in milk, especially for the Normandy breed at the very end of the lactation. Dhankhar et al. [53] proposed a method to study the influence of season on the variability of sterols in different species’ origins. Buffalo milk has a very different sterol profile compared to other animal species. In addition, seasonal variation affected especially cholesterol content compared to other minor sterols, and winter milk had a lower level of cholesterol compared to other seasons. The authors commented that the variation based on season was not able to be satisfactorily explained by PCA. However, HCA correctly grouped the 4 species of animals into 4 clusters by the sterol content. Squared Euclidean distance between objects was applied in HCA, to give the natural grouping of samples. The HCA dendrogram allowed the visualization of the similarity or dissimilarity of the measurements in 2D.

As can be observed, HCA is the main representative of the clustering methods. It is also important to note that after CA, most of the studies presented above proceeded to classification and/or regression analysis, which are presented in the next sections of this paper. Overall, in the aforementioned-studies, CA was used as a step to visualize the samples in clusters and to understand the interrelationships of the samples’ datasets, before proceeding to supervised methods.

### 2.2. Classification

The capability to assign an object to a class on the basis of its characteristics belongs to the pattern recognition field. There are many methods to classify objects and one of the applications of chemometrics is the classification of objects in groups depending on their characteristics expressed as results of a set of measurements [54]. Classification methods could be distinguished into “discriminant” and “class-modeling” techniques (Table 2).

In the first case, the technique tries to discriminate among the object’s groups dividing the model hyperspace into several regions equal to the number of classes and assigning each object to a specific region of the hyperspace on the base of its characteristics. In this way, each sample may belong to just one class. In the case of class modeling instead, the technique tries to model the analogies between objects of a class rather than observe the differences. So, each group of objects is modeled separately, and, at the end, an object could be assigned to one or more classes, or rejected as non-included in none of the classes (Figure 1).

In the discriminant classification, some methods may be counted: kNN, PLS-DA, LDA, and QDA. Instead, class-modeling techniques may be included: SIMCA, DD-SIMCA, and UNEQ [55].

Describing the details of all classification methods is out of the scope of this work, and here we will consider only the most used techniques (discriminant or class-modeling) applied to the milk and dairy product classification in milk adulteration in the last years.

A basic distinction between supervised and unsupervised classification techniques will be maintained. Supervised classification methods require some knowledge “a priori” of the classes and the method to assign or not assign samples to a certain class; in contrast, the unsupervised methods just classify samples on the base of their characteristics [56].

In recent years, the number of studies that use chemometrics to properly elaborate and interpret analytical results is largely increasing. The power of the chemometric technique is evident in all the cases where the output of an instrumental analytical technique is a spectrum, like in visible and/or infrared spectroscopy (VIS, VIS-NIRS, NIRS), nuclear magnetic resonance (NMR), or spectrometry (CG-MS, LC-MS).

Regarding classification used in milk adulteration, in the last five years, there have been several examples that used chemometrics and in Table 3 some relevant examples have been reported.

The use of chemometrics on instrumental data requires some preliminary steps, like data pre-processing or data dimension reduction. A short illustration of these steps has been reported below. In general, the application of a specific classification technique in place of another one depends on the data structure. In some cases, using one method rather than another one leads to the same results; in others, the application of a specific method could improve classification efficiency.

The classification statistical techniques most used in the last years for milk applications were PLS-DA as a pure classification technique and SIMCA as a class-modeling approach. Kamboj [57], for example, used PLS-DA to detect water adulteration in milk from NIRS spectra. Chung [58], working on isotope ratio data, used OPLS-DA to perform classification. The paper did not extensively explain the reason for this choice. Jin [59] used the least squares support vector machine (LS-SVM) for qualitative analysis of adulterated milk identification using 2D autocorrelation spectroscopic data. Karunathilaka [60] used Raman spectroscopy data from two different instruments and SIMCA for not-target classification to detect milk powder adulteration. Galvan [61], on data coming from low-cost spectroscopic devices (NIR and energy dispersive X-ray fluorescence—EDXRF), used more than one technique: PLS-DA for the EDXRF data and C-support vector classification (C-SVC) for NIR data. In the end, they concluded that DD-SIMCA was more useful to classify the samples with good accuracy (98.9%). Other two interesting uses of PLS-DA applied to NIR data were conducted by Ejeahalaka et al. [62] on cow’s milk and by Di Donato et al. [63] on donkey’s milk. DD-SIMCA is a one-class classification algorithm proposed in 2017 by Zontov [64]. The algorithm in the first phase is similar to the SIMCA algorithm, with a preliminary PCA. Then the PCA results were used to calculate the orthogonal distance and score distance for each object. These distances were then used to individuate a threshold limit value of the classification area. New samples were then classified in the orthogonal vs. score plot and assigned to the class when under the acceptance area defined for a given alpha value. Wang [65] evaluated four different classification methods (RF, LDA, SVM, and kNN) when dealing with milk authentication by infrared spectroscopy. To evaluate the best algorithm, the means of precision, accuracy, recall (true positive divided by the sum of true positive and false negative), and another parameter F1 (that together evaluate precision and recall) were calculated for each performance evaluation of all classes and for every classifier. The results indicate that RF had the best performance. In a work about image analysis [66] applied to recognize goat’s milk (as a target class) from other milk species adulterants, two methods were tested: OC-PLS and DD-SIMCA. In this case, OC-PLS was not recommended and DD-SIMCA was preferred. Chen [67] used ELM and extreme ELM (EELM) to classify six types of milk of different brands analyzed by NIRS. ELM is a regression and classification algorithm. It is simple and efficient and extremely fast. Vargas [56] applied PLS on the voltammetric characterization of fresh cow’s milk and from milk powder, using as Y the percentage of adulteration with reconstituted milk. Potocnik [68] in his paper used DA and OPLS-DA to elaborate data from isotopic ratios on types of milk to verify their geographical origins. Similarly, Xie [69] performed similar work on geographical discrimination of milk from Mongolia using isotope ratio, elements, and amino acids composition. In this paper, the chemometric analysis was performed with OPLS-DA. Tommasini [70], again using NMR, in this case, to classify the breed of cow, used PLS-DA analysis to distinguish between milk from different cow breeds, Friesian vs. autochthonous. PLS-DA and OPLS-DA, together with HCA and RF, were also cited by Sundelkide [71] to elaborate on the NMR spectra acquired in order to underline the importance and potentiality of the milk metabolomics studies. Segato et al. also used NMR to discriminate the metabolic profiles of different pasture-based alpine Asiago PDO cheeses [72]. To conclude the NMR overview, Yanibada [73] reported the application of OPLS-DA, preceded by an explorative PCA, to classify two groups of cows by NMR metabolomics. In Table 3 a synthesis of the more relevant papers identified has been reported.

To summarize, excluding PCA (mainly used to preliminarily study the problem), PLS-DA and OPLS-DA were the most used methods for classification in the recent papers on milk classification. The second most used have been SIMCA and DD-SIMCA, followed by many other various methods. The use of some classification techniques more than others could be attributed to different reasons: PLS-DA and OPLS-DA, the more used in the reviewed articles, are more known compared to some other more specific methods. The main reason for their popularity is probably linked to the fact that they are implemented in a lot of user-friendly commercial software, mainly used by non-expert users. It is advisable to use PLS-DA in place of LDA when the number of variables is higher than the number of samples and when the predictors are correlated. When classes are not balanced (i.e., the number of samples for each class is very different), better results are often obtained by class-modeling techniques, such as SIMCA. The choice of the proper classification method should also be influenced by their parametric or non-parametric nature: the former, such as LDA, assumes that the data follow a particular statistical distribution, so the model calculation becomes the calculation of the parameters of these distributions. The disadvantage of parametric techniques is that they can lead to big mistakes when starting assumptions fail to be verified. The advantage is that they make it easier to obtain the probability of obtaining a correct classification. On the other hand, non-parametric methods do not explicitly assume no statistical distribution (e.g., SIMCA, kNN, etc.).

**Table 3 foods-12-00139-t003:** Recent studies (since 2018) involving classification methods related to milk adulteration.

Type of Milk	Target	Analytical Method(s)	Classification Method(s)	Reference
Cow	Classification	NIRS	EELM	Chen [67]
Cow	Organic milk geographical indication	Isotope ratio	OPLS-DA	Chung [58]
Cow	Authenticity	NMR	CDA	Segato [72]
Goat	Adulteration detection	Image analysis	OC-Classifier, OC-PLS, DD-SIMCA	dos Santos Pereira [66]
Cow	Quality	Chemical analysis, NIRS	PCA, SIMCA, PLS-DA	Ejeahalaka [62]
Various	Authenticity	NIRS, EDXRF	DD-SIMCA, PLS-DA, C-SVC	Galvan [61]
Cow	Adulteration	IR	LS-SVM	Jin [59]
Cow	Adulteration	NIRS	PCA, PLS	Kamboj [57]
Milk powder	Adulteration	Raman	PCA, SIMCA	Karunathilaka [60]
Cow	Geographical origin	Isotope ratio	ANOVA, DA, OPLS-DA, DD-SIMCA	Potočnik [68]
Cow	Authentication	Chemical analysis	PCA, OPLS-DA	Vargas [56]
Cow	Authentication	FTIR	PCA, kNN, SVM, RF, LDA	Wang [65]
Cow	Traceability	Chemical analysis, isotope ratio,	PCA, OPLS-DA	Xie [69]
Cow	Quality, breed classification	NMR	PLS, PLS-DA	Tomassini [70]
Cow	Quality	NMR	PCA, PLS-DA, OPLS-DA, HCA, RF	Sundekilde [71]
Cow	Quality	NMR	PCA, OPLS-DA	Yanibada [73]
Donkey	Authentication	NIRS	PLS-DA, VSN, ASCA	Di Donato [63]

Abbreviations: ANOVA = analysis of variance, ASCA = ANOVA simultaneous component analysis, CDA = canonical discriminant analysis, C-SVC = C-classification support vector classifier, DA = discriminant analysis, DD-SIMCA = data-driven soft independent modeling of class analogy, EELM = ensemble of extreme learning machine, HCA = hierarchical cluster analysis, k-NN = k-nearest neighbors, LS-SVM = least squares support vector machine, LDA = linear discriminant analysis, OC = one-class classifier, OC-PLS = one-class partial least Squares, OPLS-DA = orthogonal partial least squares-discriminant analysis, PCA = principal component analysis, PLS = partial least squares, PLS-DA = partial least squares-discriminant analysis, RF = random forest, SVM = support vector machine, VSN = variable sorting for normalization.

### 2.3. Regression

Multivariate regression is widely used to quantify the concentration of adulterants in food matrices. In Table 4, the papers presented for this review in the last five years, with reference to regression methods, are listed.

The most popular multivariate regression method is certainly partial least squares (PLS) [74], as it is relatively simple to use and is implemented in a lot of statistical software, including instruments software (e.g., Opus). For this reason, in the last five years, PLS regression was used in more than three-quarters of the works on milk adulteration. The main advantage of PLS is its ability to handle data with many more variables than samples, specifically when these variables co-vary. The algorithm performs a simultaneous decomposition of both X (descriptors matrix) and Y (response matrix) matrices with the aim to maximize the covariance between the two matrices, computing at the same time latent variables (LVs) that explain the maximum variability of X. Due to its features, PLS is often used to treat spectral data, especially in the infrared region. In fact, with respect to other methods, such as chromatography, near- and mid-infrared spectroscopies (NIR and MIR, respectively) offer numerous practical advantages: they are fast, non-destructive, non-invasive, and relatively cheap techniques. Moreover, sample preparation is usually absent or extremely simple. The only drawback is the complex interpretation of the spectra, especially for NIR spectra, where differences in overtones and combination bands are difficult to detect and interpret. For this reason, the use of a simple multivariate tool for the extraction of relevant information is essential.

NIR spectroscopy is used to detect and quantify different kinds of adulterants: the most common and simple ones, such as water [57], urea [75,76,77], melamine [76,77,78], and sugar [79], and less common ones, such as sodium dodecyl sulfate (a milk surfactant) [80] or different vegetable oils added to yogurt [81]. Moreover, NIR spectroscopy is also used to detect specific adulterants for particular matrices as showed by Pandiselvam et al., where coconut milk residue was used to adulterate desiccated coconut powder [82], or by Di Donato et al., which used cow’s milk as an adulterant in goat’s milk samples [63].

MIR spectroscopy is also widely used coupled with PLS regression to detect and quantify adulterants in different milk samples. In several works, MIR was used to quantify the amount of cow’s milk in more expensive milk types: buffalo [83,84], goat [85], and horse [86]. It was used to analyze coconut milk samples adulterated with water [87]. MIR spectrometers equipped with an ATR cell were employed to detect soya bean oil and common sugar [88], sucrose [89], and formalin [13] in cow’s milk. The use of an ATR cell allows for minimizing sample preparation, as the penetration depth in the sample of IR radiation does not depend on sample thickness. Obviously, NIR and MIR spectra have to be properly pre-processed to minimize noise, scattering, and other undesirable contributions. Hence, it is good practice to build PLS models applying different combinations of pre-processing methods and compare the results to see which one provides the best prediction performance. For instance, Temizkan et al. [81] tried different preprocessing options: normalization, smoothing, first derivative, second derivative, multiplicative scatter correction (MSC), and standard normal variate (SNV). These, together with the baseline correction, are the most common row pre-processing method used to treat NIR and MIR spectra.

Another spectroscopic technique coupled with PLS in the milk adulteration field is Raman spectroscopy, whose spectra rely on the light scattering of vibrating molecules. Raman spectroscopy was employed to find maltodextrin, sodium carbonate, and whey in bovine milk [90,91], as well as margarine, palm oil, and corn oil in cheeses made using adulterated milk samples [92,93].

Although in the majority of papers PLS regression is applied to vibrational spectroscopic data, in recent literature, there are also many applications with different techniques. Cyclic voltammetry, using a graphite/SiO_2_ hybrid-working electrode, was employed to quantify reconstituted skim milk in cow’s milk [11], electrochemical impedance spectroscopy was used to measure urea [36] whereas face fluorescence spectroscopy and laser-induced breakdown spectroscopy assessed the amount of bovine milk in buffalo milk [90] and ovine and caprine milk [94], respectively. Moreover, time-domain NMR [12] and opto-electronic nose [40] quantified formaldehyde in bovine milk. The versatility of this technique is one of the reasons why its presence is predominant among papers that deal with multivariate regression. Actually, in many papers, PLS is frequently compared with other two multivariate regression methods, i.e., multiple linear regression (MLR) [95] and principal component regression (PCR) [96]. Jaiswal et al. [85] and Gonçalves et al. [84] showed comparable results between PLS and MLR in quantifying adulterants with MIR spectroscopy. Conceição et al. [97] used MLR coupled with MIR spectroscopy to assess the amount of sodium bicarbonate, sodium hydroxide, hydrogen peroxide, starch, sucrose, and urea in cow’s milk. However, the use of MLR is not recommended if the data matrix is ill-conditioned, namely has more variables (e.g., wavenumbers) than samples, and if those variables co-vary, as the regression model would be unstable. On the other hand, PCR is a more reliable method, since the variables are orthogonal (the ill-conditioned matrix problem has been overcome) and only relevant information in the original data matrix is considered, being based on PCA. Unlike PLS, in PCR the information in the response matrix (Y) is not taken into account when choosing the number of PCs. Moreover, for this reason, PLS has been habitually preferred to PCR. In some of the papers inspected for this review, these two methods were compared: on three occasions PLS provided the best prediction performances [13,86,89], whereas in one case the results obtained by the two methods were similar [87].

Throughout the years, the PLS algorithm has been modified by many authors to add features and make it more suitable for specific tasks (e.g., multiblock analysis, locally weighted models, etc.). One of the most famous extensions of PLS is orthogonal PLS (OPLS) [98], which removes the systematic variation from X that is not correlated (orthogonal) to Y. It was used by Delatour et al. [99] on data collected from eight different NIR and MIR miniature sensors to measure the amount of semicarbazide hydrochloride, ammonium sulfate, and cornstarch in skimmed milk powder [96]. Another different use of PLS regression, synergy interval PLS (siPLS) [100], has been used by Vinciguerra et al. to quantify cheese whey in cow’s milk samples through MIR spectroscopy [38]. In this method, the MIR spectra were divided into different intervals (8, 16, 32, 64, and 128) with the same number of variables, applying a PLS on each interval. Furthermore, combinations of these intervals (two by two, three by three, and four by four) were also explored and PLS was performed for each combination. Hosseini et al. used the genetic algorithm PLS (GA-PLS) in order to perform an efficient variable selection before calculating the regression models [80]. Lastly, unfolded PLS with residual bilinearization (U-PLS/RBL) [101] coupled with fluorescence spectroscopy was used by Barreto et al. to quantify melamine in bovine milk [102]. Actually, U-PLS/RBL belongs to the family of multiway methods, similar to other techniques such as parallel factor analysis (PARAFAC) and multivariate curve resolution-alternating least squares (MCR-ALS), all based on obtaining pure profiles of the components present in a mixture system. They are also called second-order calibration algorithms, as they can operate by decomposing the 3-way data matrix and then performing a regression between the resolved relative concentration of the constituents of interest and the corresponding reference concentration. Fluorescence spectroscopy provides excitation-emission matrices (EEMs) that can be resolved by those algorithms. According to de Araújo Gomes et al., U-PLS/RBL is particularly suitable to deal with fluorescence data, as it is able to model the inner filter effect that occurs in chemical fluorescence spectroscopy analysis systems [103]. Barreto et al. also used PARAFAC to quantify melamine, obtaining slightly better results than the ones achieved with U-PLS/RBL. PARAFAC [104] is a generalization of PCA to higher-order matrices, and its models furnish parameters (loadings) that describe the variability in the samples. Hence, MCR-ALS [105] was used by Zhao et al. on NIR data to compute calibration models for the simultaneous quantification of multiple adulterants (urea, melamine, and starch) [77]. In this case, MCR-ALS was used on classical 2-way data (i.e., NIR spectra), but the assumptions made earlier are valid. In general, MCR decomposes the data matrix into a bilinear model constraining the components’ profiles to assure that the solution makes sense not only from a statistical point of view, but also chemically. ALS optimization explores the possible solutions through an iterative least square calculation until convergence is achieved.

Moving forward, some other less popular (but no worse) applications of multivariate regression techniques employed in the area of milk adulteration than PLS and its extensions can be found in the literature. Artificial neural network (ANN) regression methods, namely generalized regression-NN [106] and back propagation-ANN [107], were used to assess the amount of melamine, wheat flour, and corn flour in milk powder samples [108] and acidity in cow’s milk samples [109], respectively, both through Raman spectroscopy. Least squares support vector machine (LS-SVM) [110] was applied on both NIR and dielectric spectroscopic data to quantify mature bovine milk in colostrum samples [111] and on MIR data to assess cheese whey in bovine milk [38], providing better results than PLS. A generalized linear model with lasso regularization (GLM-Lasso) [112] coupled with MALDI-TOF mass spectroscopy provides better results than PLS too, in this case, to detect bovine milk in caprine and ovine milk [113]. Ehsani et al. applied boosted regression tree (BRT) [114] on NIR spectra collected by a portable spectrometer for a fast water quantification in cow’s milk [115]. The presence of water in cow’s milk was also inspected by Asefa et al. [116], who proposed a procedure based on digital image analysis coupled with extreme gradient boosting (XGBoost) [117].

To sum up, the most-used technique for multivariate regression in the field of milk adulteration is by far PLS, as it is relatively simple to use and is present in much commercial software. In most cases, proper use of PLS regression is enough to obtain good prediction performances, but in the case of a more complex data structure, it is worth trying more advanced techniques. The use of the many extensions of PLS can be useful to increase the signal-to-noise ratio, to compute prediction models only with the most relevant variables, or to deal with 3-way data. More expert users sometimes use other kinds of multivariate regression methods, such as ANN or SVM. In some cases, they provide slightly better results than PLS, but in many other cases, the results are comparable.

**Table 4 foods-12-00139-t004:** Recent studies (2018–2022) involving regression methods related to milk adulteration.

Type of Milk	Target	Analytical Method(s)	Regression Method(s)	Reference
Cow milk	Water	NIR	PLS	[57]
Cow milk	Urea	NIR	PLS	[75]
Fat-filled milk powder	Melamine, urea	NIR	PLS	[76]
Goat milk powder	Melamine, urea, starch	NIR	PLS, MCR-ALS	[77]
Milk powder—infant formula	Melamine, vanillin	NIR HSI	PLS	[78]
Cow milk	Sugar	NIR	PLS	[79]
Cow milk	Anionic surfactant (SDS)	NIR, MIR (ATR)	PLS, GA-PLS	[80]
Yogurt	Margarine, sunflower oil, corn oil, hydrogenated vegetable oil	NIR, MIR	PLS	[81]
Desiccated coconut powder	Coconut milk	Vis-NIR	PLS	[82]
Donkey milk	Cow milk	NIR	PLS	[73]
Buffalo milk	Cow milk	MIR	PLS	[83]
Buffalo milk	Cow milk	MIR	PLS, MLR	[84]
Goat milk	Cow milk	MIR, Raman	PLS	[85]
Horse milk	Cow milk, goat milk	MIR	PLS, PCR	[86]
Coconut milk	Water	MIR	PLS, PCR	[87]
Cow milk	Soya bean oil, sugar	MIR (ATR)	PLS, MLR	[88]
Cow milk	Sucrose	MIR (ATR)	PLS, PCR	[89]
Cow milk	Formalin	MIR (ATR)	PLS, PCR	[13]
Cow milk	Maltodextrin, sodium carbonate, whey	Raman	PLS	[90]
Cow milk	Whey	Raman	PLS	[91]
White ultra-filtered cheese	Margarine, palm oil, and corn oil	Raman	PLS	[92]
Cow milk	Reconstituted skim milk powder	Cyclic voltammetry	PLS	[11]
Cow milk	Urea	Electrochemical impedance spectroscopy	PLS	[36]
Buffalo milk	Cow milk	Face fluorescence spectroscopy	PLS	[93]
Ovine and caprine milk	Cow milk	Laser-induced breakdown spectroscopy	PLS	[94]
Cow milk	Formaldehyde	TD-NMR	PLS	[12]
Cow milk	Formaldehyde	Opto-electronic nose	PLS	[40]
Cow milk	Sodium bicarbonate, sodium hydroxide, hydrogen peroxide, starch, sucrose, urea	MIR (ATR)	MLR	[97]
Skimmed milk powder	Semicarbazide hydrochloride, ammonium sulfate, cornstarch	NIR (miniature spectral devices)	OPLS	[99]
Cow milk	Whey	MIR	PLS, siPLS, LS-SVM	[38]
Cow milk	Melamine	Fluorescence spectroscopy	PARAFAC, U-PLS/RBL	[102]
Milk powder	Melamine, wheat flour, corn flour	Raman	GRNN	[108]
Cow milk	Acidity	Raman	PLS, BP-ANN	[109]
Colostrum	Mature cow milk	NIR, dielectric spectroscopy	PLS, LS-SVM	[111]
Ovine milk and caprine milk	Cow milk	MALDI-TOF-MS	PLS, GLM-Lasso	[113]
Cow milk	Water	NIR (portable)	BRT	[115]
Cow milk	Water	Digital image analysis	XGBoost	[116]

Abbreviations: ATR = attenuated total reflection, BP-ANN = back propagation artificial neural networks, BRT = boosted regression trees, GA-PLS = genetic-algorithm partial least squares, GLM-Lasso = generalized linear model with lasso regularization, GR-NN = generalized regression neural networks, HSI = hyperspectral imaging, LS-SVM = least squares support vector machine, MALDI-TOF-MS = matrix-assisted laser desorption ionization time-of-flight mass spectrometry, MCR-ALS = multivariate curve resolution alternating least squares, MIR = mid-infrared, MLR = multiple linear regression, NIR = near-infrared, OPLS = orthogonal partial least squares, PARAFAC = parallel factor analysis, PCR = principal component regression, PLS = partial least squares, siPLS = synergy interval partial least squares, TD-NMR = time-domain nuclear magnetic resonance, U-PLS/RBL = unfolded partial least squares with residual bilinearization, Vis = visible, XGBoost = extreme gradient boosting.

## 3. Steps for Development and Validation of a Chemometric Approach

It is difficult to define a precise pipeline for the correct development and validation of a chemometric approach for authentication purposes. This chapter tries to face the fundamental steps, covering the sampling procedure, considering the analytical source of data, the model calibration and validation, and the main figure of merits useful for model evaluation (Figure 2).

### 3.1. Correct Sampling Procedure

#### 3.1.1. Sampling Strategies

No matter the chemometric model to be performed, according to the developed strategy goal, it is mandatory to perform a proper sampling strategy. Behind the word “proper” there are a set of extremely challenging standpoints that should consider the nature of the sample, the statistical representativeness, the analytical chemistry principles, and the quality and the management of the obtained datasets. Sampling procedures are very important to assure the robustness and reliability of the developed chemometric models. However, no well-defined sampling protocols exist so far for fingerprint techniques.

When addressing the nature of the sample, a relevant emphasis should be placed on the heterogeneity of a food matrix, together with the wide possibility of frauds, from the adulteration, i.e., the presence of an undeclared substance to the geographical origin discrimination, passing through the substitution of ingredients or commodities. In any case, the source of the samples, i.e., the provider, must be extremely reliable when addressing an authentication issue. They must be of provable provenance to assure they are authentic or not; thus, it would be advisable to obtain them from the producer rather than buying at retail markets [21].

For instance, the collection of commercial samples from local grocery stores to study goat’s milk adulteration by cow’s milk [85] could be inappropriate. Indeed, the commercial milk already passed to technological operation (heat treatments, fat separation, homogenization); thus, it would be more representative of real fraud to mix the different types of milk before any unit operation. This is what was done by Spina et al. [83], who described in detail the farmers, the breeds, and the sampling period and batches. Furthermore, they strengthened their experimental plan by planning a randomized pairing of cow and buffalo milk to obtain 17 adulteration levels.

Pandiselvam et al. [82] also adopted the strategy of ad hoc sample preparation. They prepared different adulterated samples by adding coconut milk residue to desiccated coconut powder. Even though the sample numerosity was quite high, i.e., 20 samples prepared for ten adulteration levels (from 0 to 100% *w/w*), it seems that the raw materials used to prepare the standard samples were always the same, thus not covering all the possible sources of variability. The variability of simulated adulterated samples was better covered by de Oliveira Mendes et al. [88], who considered six samples of milk from different producers to be adulterated with sweet whey prepared at a laboratory scale at eight adulteration levels.

From a statistical standpoint, the size and the representativeness of the sample collection must be considered [32] to obtain samples spanning all the sources of variability associated with the application of the model [118]. Different strategies described by the theory of sampling (ToS) could be followed to guarantee representative sampling and appropriate analytical quality [119]. A power analysis could be performed to establish the adequate number of samples required and to reduce the technical and biological variability. When a wide variability should be covered in a limited set of measures, design of experiments (DoE) techniques could be applied to obtain statistically valid data; the advantages of these approaches are well described by Peris-Díaz & Krężel [120].

In the literature there are examples of poor sampling strategies; for example, there are works considering a number of samples that is too low to be representative from both a technological/chemical and statistical point of view [12,38,87].

From an analytical point of view, the sample handling in terms of conservation prior to analysis, preparation, and analytical replicates should be faced to circumscribe the intrinsic variability. This is quite a challenging issue which has been clearly pointed out by Kemsley, et al. [121], and too often poorly described in the revised literature.

Finally, to sum up the useful sampling strategy to be adopted, the approach proposed by the “Five Ws” iterative interrogative technique could be winning. The first W to be clearly set is the goal of the developed approach, i.e., why, and the definition of the authentication issue to be addressed. Then, it is appropriate to cover the personnel and instrument variability (who), together with the definition of sample unit, the number of samples, handling procedure, representativeness, balanced/not balanced datasets, and possible development/availability of trusted samples (what). Moreover, the range of time (when)—which could refer to seasons, harvesting years, vintages, product aging, and so on—should be adequately covered. Finally, the investigation of the effects of the area of origin and/or the processing steps (where) should be faced.

#### 3.1.2. Data Quality

The quality of the collected raw data strongly influences the data processing and the model quality. This is highly dependent on the instrumentation characteristics and related analytical methodology. The review by Szymanska [122] deeply described the four main dimensions of data quality (accuracy, completeness, timeliness, and consistency) and their characteristics. The most common artifacts generated by quality collection failures are missing values, outliers, noise, and misalignments. According to the type, there are strategies for their detection and deletion, substitution, or correction [122]. However, in most of the literature, little attention is given to the description of these strategies, which are hopefully applied to assess and monitor the quality of the collected data before the chemometric model construction.

### 3.2. Pre-Processing

An exception is the description of data pre-processing, which is generally reported as a winning strategy to remove irrelevant sources of variation, such as instrumental and experimental artifacts due to the employed analytical method. However, there are still authors who miss the preprocessing description, such as Kamboj et al. (2020) [57], or just mention an automatic strategy applied by the software. Different preprocessing strategies are available; in-depth information is given by Engel et al. [123]. Every specific dataset has specific features; thus, the definition of a rule of thumb to define which preprocessing strategy is more appropriate is impossible.

In any case, the spectroscopic data requires a pre-processing step before the statistical data analysis to remove or minimize variability in the spectra not related to the sample’s characteristics. It will be clear that pre-processing cannot generate information, but only help to extract proper information already existing in the data. Moreover, incorrect use of pre-processing may cause a loss of information. Pre-treatment should be well calibrated to minimize the effects of “noise” such as optical phenomena, effects of temperature changes, light scattering, baseline shift or trends, and so on.

Most of the revised works, especially the ones dealing with infrared data, apply different preprocessing strategies, such as smoothing, standard normal variate (SNV) or multiplicative scatter correction (MSC), and derivatives alone or in combination [87,99,124]. Later on, they select the most appropriate one to solve the specific adulteration issue based on the performance criteria obtained in the developed models. However, it is important not to apply all of them by default without looking back at their effect on the data. Indeed, it should be considered that an inappropriate transformation can cause alterations to data quality, driving relevant consequences on model outcomes. A must-read tutorial concerning pre-processing has been written by Oliveri et al. [125].

Between the papers explored, some different approaches have been found in NIR pre-processing. Ejeahalaka [62] performed a comparison between two different approaches: first, no pre-processing at all, and second, extended multiplicative signal correction (EMSC) on a selected part of the spectrum. In Galvan [61] some different pre-processing methods were tested before a mean centering for all: (1) raw data, (2) Savitzky–Golay smoothing (third-order polynomial and 21 window points), (3) standard normal variate (SNV), (4) multiplicative scatter correction (MSC), (5) first and second derivative with Savitzky–Golay smoothing, (6) SNV plus first and second derivative, and (7) MSC plus first and second derivative. At the end, the best performance (evaluated by RMSE of the calculated models) was obtained by the application of the first derivative with smoothing (pre-processing 5). Wang [65] used three pre-processing steps: (1) mean centering, (2) first, and (3) second-order Savitzky–Golay derivative, selecting at the end the first-order derivative as the better pre-processing method.

Kamboj [57] did not indicate which pre-processing was used. Not mentioning the pre-processing step should be avoided because this step implies some assumptions on the nature of the data set variability, and it is crucial that these assumptions are well understood and appropriate. An innovative approach was reported by Di Donato [63] in a study on donkey milk. NIR data were used to identify and quantify cow adulteration in expensive donkey milk. In this case, the pre-processing was done by variable sorting for normalization (VSN), a recent scatter correction technique [126] that estimates the weight of wavelengths that are or are not related to scattering effects instead of that related to the response of interest. Not-related wavelengths were not considered in the successive step. In this way, it is possible to obtain an improvement in signal and model interpretation.

Karunathilaka [60] in an application of Raman spectroscopy cites different spectral pre-processing to remove fluorescence and laser fluctuations, including Savitzky–Golay first and second derivatives and standard normal variate (SNV), choosing at the end the second derivative.

### 3.3. Data Reduction

The analysis of spectroscopic results is a typical example in which the dimension of the analytical part of the dataset (n columns) is much higher than the number of samples (m rows), normally thousands of columns vs. tens or hundreds of rows. So, to avoid elaboration problems and to select just the variables relevant for the statistical analysis, a variable selection step is often evaluated. Reviewing in detail all the possible algorithms is out of scope, considering their relevant number; thus, here we only report the ones used in the evaluated papers. Between them, just a few used a data reduction algorithm. For example, Chen [67] on NIRS data used an extension of the ReliefF filter algorithm [74]. ReliefF filter works on multiple classes, building a weight vector that indicates for each feature (wavelengths in the NIRS case) how important it is to explain the differences between samples of different classes. Wang [65] instead used just an observation of the first two PCA loadings as the criterion to understand relevant wavelengths, but it was unclear if just the relevant wavelengths in the subsequent classification step were used.

### 3.4. Use of Robust Validation Procedures

Before detailing the possible validation procedures, it is essential to consider the quality of the calibration. Taking for granted that the data representativeness and numerosity must be guaranteed according to the defined purpose, it is relevant not to overfit or underfit the model calibration.

Model validation is frequently addressed by iterative validation procedures, such as cross-validation. In the considered papers, the most used cross-validation strategy is leave-one-out, to whatever degree it should be avoided for its over-optimistic results, especially in the case of exhaustive sampling procedures [13,75,83]. Indeed, it means that during the iterative recalculation of the model just one sample at a time is removed; this way the robustness of the model is poorly investigated. None of the work internally validates models with other iterative procedures such as Monte Carlo, Jackknife, holdout, or bootstrapping.

The use of internal validation is often justified when a low number of samples is at disposal. In these situations, it can be unaffordable to exclude 30–40% of the collected data to be used as a test set. Westad and Marini [127] suggest this strategy when the number of samples is smaller than 40.

Moreover, the internal validation procedures are fundamental insights to study the model stability, identify the main sources of variation, and improve model performance, i.e., by setting model dimensionality [128]. This was the approach followed by Ejeahalaka et al. [76] for both SIMCA and PLS model development. It is important to notice that the correct model dimensionality is fundamental for predicting the test set; if the model dimensionality is incorrect, the performance criteria/figure of merit may not be a good estimate of future samples, as reported by Westad and Marini [127]. For instance, the results obtained from internal validation give insights about model overfitting due to the selection of a huge number of components/variables, which means fitting too much of the data so that also the measurement noise is interpreted as a relevant effect.

Then, it is the time to use robust, mandatory validation procedures in order to guarantee reliable and reproducible results. Usually, the available samples are divided into two subsets: a training (or calibration) set to be used for building the model, and a test set used to evaluate its validity [20] in terms of quality and generalization ability [129]. The division should guarantee that the calibration set covers the whole variability domain to obtain reliable results. The dataset split could be performed arbitrarily—according to the acquired knowledge of the data, randomly, or designed by sampling strategies—such as the Kennard and Stone algorithm, Duplex, D-optimality criterion, and K-means or Kohonen mapping; for more details about the differences among the strategies and their effects refer to Westad and Marini [127].

Infrequently, the experimental structure is considered for data splitting. This was the case for Genis et al. [92] who considered 15 concentrations of fat in the calibration set, and 11 concentrations of fat as validation data set when developing methods for the identification of foreign lipid types and adulteration ratio in milk. Most of the revised papers apply random sample selection to build the test set considering from 40 to 20% of the whole data. Among the designed sampling strategies, the Kennard and Stone algorithm is the one mostly used. However, in many cases no information is provided for dataset splitting, thus making the model robustness evaluation difficult.

In any case, it would be advisable to use a fully independent set of data to test the model; for example, considering a different production batch, a different time of the year, or a different harvesting year.

This option will represent the ideal procedure for model validation, anyway it should be set to guarantee the samples’ diversity if possible, or at least their mutual independence [130].

If someone argues it is still not enough, we can reply as suggested by Westad and Marini [127]: “Another way to overcome the problem of using the same criterion to select a subset of variables and the error (i.e., cross-validation) is to divide the objects into a calibration, a validation and a verification set, where the verification set is the ‘proof of the pudding’”.

Each step of model development (i.e., calibration, cross-validation, and external validation) should be properly evaluated by diagnostic metrics (i.e., Figures of Merits), which are discussed in the next session.

### 3.5. Performance Criteria/Figure of Merits

Before mentioning the performance criteria useful for regression evaluation, it is important to have enough information to evaluate the quality of the collected data. In particular specific information must be reported about the numerosity of the data, their variability (i.e., mean, median, and standard deviation), the nature of the measure (instrumentation used), the removed outliers (and adopted strategy), the regression algorithm employed (mainly PLS, OPLS, PCR, MLR, LSSVM, SWM, ANN, GLM-Lasso, and so on), or the classification approach (mainly PLS-DA and OPLS-DA for the pure classification, and SIMCA and DD-SIMCA for the class-modeling techniques), the characteristics of the model development steps (calibration, internal- and external validation), the potential data pre-treatments, and the selected components/latent variables [131]. Last but not least, the information about the reference method employed to determine the specific compound and the associated error, i.e., the standard error of the laboratory (SEL), or the standard error of the test (SET), must be reported [131]. Having a clear idea of the variance covered by the data and the error of the reference analysis would be crucial to judge the results obtained by the regression model obtained. Indeed, the accuracy of chemometric model predictors depends on the repeatability of the reference methods and it combines both the error of the reference measure and the error of the fingerprint analysis [132].

#### 3.5.1. R^2^ (Coefficient of Determination) and RMSE (Root Mean Squared Error)

The main effective tests used to evaluate multivariate regression models are R^2^, SEP, and the RPD. R^2^, the coefficient of determination, is commonly used to evaluate regression models in every development step. It is quite relevant to compare the different coefficients of determination obtained in calibration, cross-validation, and prediction to understand the model stability. It would be better to evaluate the R^2^ adjusted, which corrects for the number of explanatory terms in relation to the number of data points.

The coefficient of determination (R^2^) is, in its most general definition, computed by:(1)$$R^{2}=1-\frac{SS_{res}}{SS_{tot}}$$
where SS_res_ is the sum of squares of residuals for measurements y_i_ and mean of observed data (Ῡ) and SS_tot_ is the total sum of squares.

The R^2^ adjusted is:(2)$$R_{adj}^{2}=1-\frac{n-1}{1-k-1}\;\frac{SS_{res}}{SS_{tot}}$$
where n is the number of observations and k is the number of independent variables.

However, the evaluation of R^2^ alone is not exhaustive: there may be models with high coefficient values, thus describing high data variability, but with high error, expressed as root mean square error. To determine the reasonability of RMSE value it should be compared to measurement errors such as reference method, reproducibility error, historical data, and so on.

The RMSE is computed as:(3)$$RMSE=\sqrt{\frac{1}{n}\overset{n}{\underset{i=1}{\sum }}{\left(y_{i}-\hat{y}\right)}^{2}}$$
where n is the number of observations, y_i_ is the predicted value and ŷ is the actual value.

If divided by the standard deviation of the experimental values it is obtained the normalized RMSE (nRMSE), which is an unbiased measurement for model predictions.

Good error estimation was performed for the models developed by Genis [92]. They calculated the relative error of standard deviation (RSD) and relative error of prediction (REP) together with the limit of detection (LOD) and the limit of quantification (LOQ) in the regression model intended for fat authenticity in milk for ultra-filtered white cheese.

The use of both criteria, R^2^ and RMSE, is relevant especially in cases of high range of variability of the considered compound; in this case, it could be plausible to obtain a model with higher R^2^, but accompanied by higher RMSE, if compared with a dataset with limited range of variability. Generally speaking, “wide” calibration could be less precise, but more dangerous is a too-narrow calibration which will be valid just for the case understudy [132].

The ratio between the SD and the RMSE is referred to as ratio percentage deviation (RPD). It can be seen as a performance criterion like R^2^, even if RPD is a ratio of SD, whereas R^2^ is a ratio of variance. Its calculation is present in few papers dealing with milk adulteration [13,82,84,89,91], but its use can give an immediate insight to evaluate the predictions as well as to compare models predicting different compounds [132]. There are different papers that give an interpretation of model performance according to RPD values, among them the one of Williams [133] which defines six levels of performance. In the considered works the RPD was always quite high. Indeed, very good prediction capabilities were reached by the MLR model for buffalo’s milk authenticity verification developed by Gonçalves et al. [84]; the RPD was 7.9. When developing a PLS regression on the same data it improved to 9.0, thus demonstrating the excellent performance of mid-infrared spectroscopy to assess cow’s milk levels in buffalo’s milk. The model developed by Pandiselvam et al. [82] for the detection of adulteration with coconut powder also achieved excellent performance, resulting in an RPD of 11. Excellent performances were found by Balan et al. [13] when developing a PLS model to predict formalin in cow’s milk, reaching an RPD above 8. Also, the RPD of the PLS models developed by Balan et al. [89] was high (13.4), demonstrating an excellent prediction capability of sucrose in milk, thus being able to detect sucrose addition intended to increase total solid content as well as the sweet taste. Similarly, de Oliveira Mendes et al. [91] developed a PLS model for whey quantification in raw milk by Raman spectroscopy obtaining an RPD of 13.9.

In any case, where RPD is not reported as a model parameter, it can be calculated directly from the R^2^ such as 1/−(1 − R^2^).

Bellon-Maurel et al. [134] proposed to substitute RPD with a new index, RPIQ (ratio of performance to IQ). The index is based on quartiles, thus better representing the population distribution. They found out that, in sample sets with skewed distribution, the RPD is not a good approach for SEP standardization according to population spread, whereas the RPIQ index, in which standard deviation is replaced by IQ (=Q3 − Q1), better considers the spread of the population. However, none of the works considered here applied this figure of merit.

#### 3.5.2. Specificity and Sensitivity, and Graphical Representations

The performance of classification models is assessed by verifying if samples belonging to the class of interest are designated as true positives (TP) or false negatives (FN), as well as if samples not belonging to the class of interest are labeled as false positives (FP) or true negatives (TN) [20]. Just to recall the theory, TP defines the samples recognized to belong to the class a priori assigned, FN are samples erroneously rejected, FP are samples erroneously assigned to the class, and TN are samples correctly refused.

From their assignments, it is possible to calculate the sensitivity and sensibility of the method. Sensitivity is the true positive rate (TPR), computed as TP/(TP + FN). Specificity is the true negative rate (TNR), computed as TN/(TN + FP).

The graphical tool used to represent the performance criteria of a discriminant model is the receiver operating characteristic (ROC) curves (Figure 3a). The plot represents a two-axis Cartesian space, with the horizontal axis reporting FPR, and the vertical axis the TPR. The dashed diagonal represents the performance of a random classifier. Two examples of classifiers (green and red) are shown, representing good and scarce results, respectively. The curves are built by connecting with a line the experimental outcomes. This tool is useful to compare the performances of models obtained with different parameter settings, such as the threshold value. A detailed analysis of ROC curves is discussed by Oliveri [20].

If discriminant methods can be applied only to solve multi-class situations, class modeling can be used to address both multi-class and one-class problems.

When performing a class-modeling analysis it could be useful to evaluate the results with a graphical representation, so Coomans’ plots (Figure 3b). In a two-class problem, the two axes represent the distances of samples from the models of Class 1 (○) and Class 2 (star), respectively. The two dashed lines correspond to the critical acceptance levels for each model at the defined confidence level (normally 95%). Samples of the two classes are projected as scatter points, with coordinates indicating the relative similarity with the two models in the four sectors defined in the plot. In sector 1 it is possible to find samples accepted only by Class 1 (○); in sector 2 it is possible to find samples accepted only by Class 2 (star). Both sectors include samples defined as TP for the a priori defined class.

In sector 3 are positioned samples accepted by both models; indeed, since models for each class are independently built, class spaces may overlap. Lastly, in sector 4 it is possible to observe samples rejected by both models, which highlights that the used variables do not completely resolve the class space. They prevent the forced (but possibly wrong) classification of samples that may occur in discriminant approaches [20].

## 4. Methods for Rapid and On-Site Detection to Combat Milk Adulteration

The dairy industry as well as regulatory bodies are looking for simple and rapid methods for the detection of milk adulteration [135]. Lateral flow immunoassays (LFIAs) have been used as in situ screening tools to monitor food raw material quality as they provide rapid results [136]. LFIAs have been developed, among other applications [137], for the detection and quantification of mycotoxins [138], such as aflatoxin M1 [139]. LFIAs have been also used for the detection of adulteration of milk with melamine [140]. In a very recent study adulteration of cow’s milk with buffalo’s milk was detected by an on-site carbon nanoparticle-based lateral flow immunoassay in 10 min, with the sensitivity of the test being 5%, i.e., 5% adulteration of cow’s milk with buffalo’s milk, proving that this tool is suitable for rapid detection of adulteration [135].

Another novel technology for the rapid detection of milk adulteration is DNAFoil. It is a portable, fully self-administered, on-site DNA test that does not require the use of expensive PCR equipment or laboratory setups to confirm the detection of milk adulteration within a short period of time. The efficiency of the DNAFoil kit used to detect the vegetable material in milk products (DNAFoil UniPlant) was confirmed using real-time PCR assays. The results showed that using 24 μL of DNAFoil UniPlant master mix, a 17.5 min reaction time allowed the detection of 10% adulteration of liquid cow’s milk by wheat flour [141].

Moreover, an electronic nose (e-nose) system is being evolved for the falsification detection of milk and dairy products in a reliable and rapid way [142]. This technology avoids the disadvantages of chromatography, spectrometry, and chemical methods with high costs and long cycle times [143]. Adulteration of bovine milk with formaldehyde, based on aldehydes and ketones, was examined by electronic nose by Mostafapour et al. [40]. In another investigation, the identification of trace amounts of detergent powder in raw milk using a customized low-cost electronic nose was achieved [144].

## 5. Conclusions

An overview of the different chemometric techniques (from clustering to classification and regression applied to several analytical data) has been presented along with spectroscopy, chromatography, and electrochemical sensors as well as rapid and on-site detection devices in the fight against milk adulteration and fraud. HCA is the main representative of the clustering methods. The classification of objects in groups depending on their characteristics expressed as results of a set of measurements is one of the applications of chemometrics. Classification methods were distinguished into “discriminant” and “class-modeling” techniques. The classification statistical techniques mostly employed in the last few years for milk applications were PLS-DA as a pure classification technique and SIMCA as a class-modeling approach. Multivariate regression is widely used to quantify the concentration of adulterants in food matrices and was deeply described.

Finally, the steps which should be followed to develop a chemometric model to face adulteration issues were carefully presented with the required critical discussion describing sampling strategies, pre-processing, data reduction, and use of robust validation procedures along with performance criteria/figure of merits.

All chemometric methods, supervised and unsupervised, had fundamental results in order to serve the goals of each research study. It cannot be concluded which chemometric method is the best, as each dataset is unique and different. Robustness is usually more related to supervised methods, but unsupervised methods are also important in the field. Usually, the availability and access to each chemometric method are the variables that influence their specific selection. With regard to the field of milk adulteration, it is clear that, in most cases, the simplest methods are enough to obtain good results. However, even the simplest methods are in some cases used improperly, making the results obtained inconsistent.

## Figures and Tables

**Figure 1 foods-12-00139-f001:**
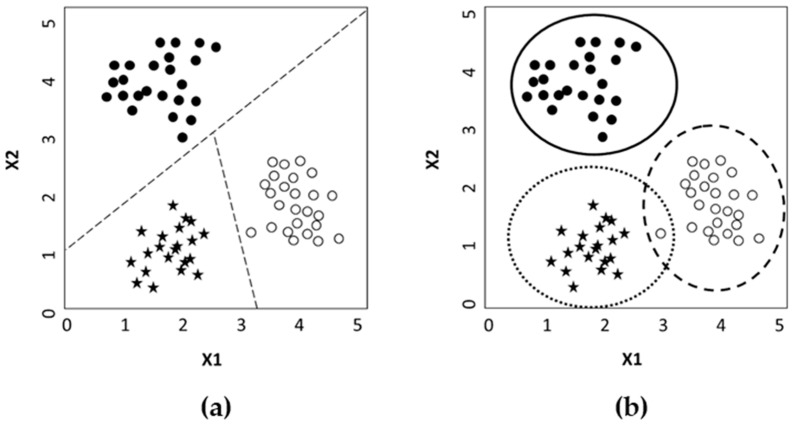
Example of difference between discriminant (**a**) modeling and (**b**) classification methods. In (**a**) the hyperspace is divided into regions equal to the category number.

**Figure 2 foods-12-00139-f002:**
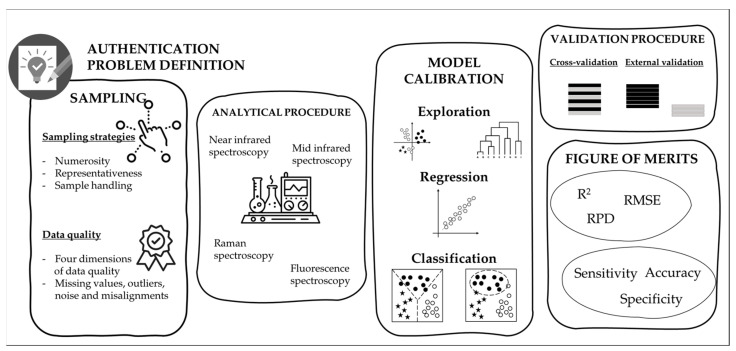
Schematic representation of the main steps useful to develop and validate a chemometric approach.

**Figure 3 foods-12-00139-f003:**
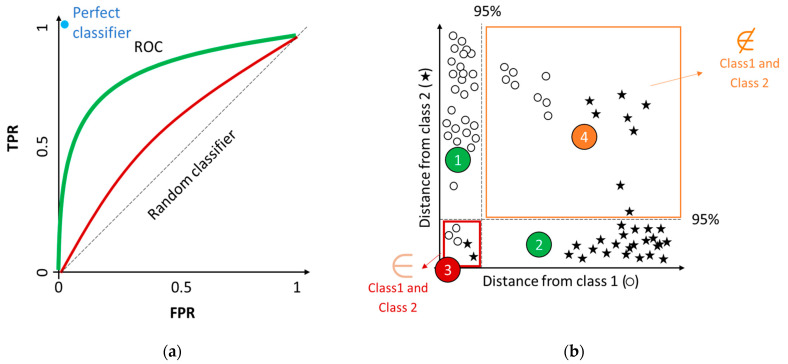
Graphical tools to represent classification model performance. (**a**) Receiver operating characteristic (ROC) curves; (**b**) Coomans’ plot.

**Table 1 foods-12-00139-t001:** Recent studies (2015–2021) related to milk adulteration and authenticity in combination with clustering analysis.

Type of Milk	Target	Analytical Method(s)	Clustering Method	Approach	Reference
Milk adulteration					
Cow’s, sheep’s, and water buffalo’s origin milk	Adulteration from different species’ origin milk	FTIR	HCA	method	[37]
Bovine milk	Adulteration with urea	EIS	HCA	Euclidean distance	[36]
UTH milk samples (skimmed and semi-skimmed) and raw milk	Adulteration with cheese whey, based on quantification of caseinomacropeptide	FTIR-ATR	HCA	Euclidean distance and Ward’s method	[38]
Cow milk	Adulteration with melamine and urea	Electrochemical biosensor	HCA	Ward’s method	[39]
Bovine milk	Adulteration with formaldehyde, based on aldehydes and ketones	Colorimetric sensor array	HCA	-	[40]
UHT whole bovine milk and UHT goat milk	Adulteration with soymilk in bovine and goat milk, as well as bovine milk in goat milk.	NMR	CA	The minimum distance method	[41]
Raw cow milk	Adulteration with Sodium Salicylate, Dextrose, Hydrogen Peroxide, Ammonium Sulphate	Sensor system	k-means clustering algorithm	-	[42]
Milk authentication					
Powder and liquid milk	Type of milk based on metal profiles	ICP-OES	HCA	Euclidean distance and Ward’s method	[43]
Organic and conventional milk	Type of milk (organic vs. conventional) based on organic status and trace element content	ICP-MS	HCA	Euclidean distance and Ward’s method	[44]
Malaysian vs. milk from other countries	Geographical origin, based on metal content	ICP-MS	HCA	Ward’s method	[45]
-	Geographical origin, isotope ratios, metals, and fatty acids	CF-IRMS (*δ* ^18^O), EA-IRMS (*δ* ^13^C and *δ*^15^N), GC (fatty acids), ICP-OES (Na, K, Mn, P, Zn, Ca, Fe, and Mg), and ICP-MS (other metals)	HCA	-	[46]
Cow milk	Geographical origin, based on stable isotope ratios	IRMS and CRDS	HCA	-	[47]
Raw milk	Geographical origin, based on stable isotope ratios and metal content	IRMS and ICP-MS	HCA and k-means clustering algorithm	HCA: Euclidean distance and Ward’s method K means: 200 iterations and 25 random starting points	[48]
Cow, goat, camel, donkey, and yak milk	Species recognition based on sn-2 and sn-1,3 fatty acid composition and sterols	GC, GC-MS	HCA	-	[49]
Fresh buffalo, bovine, and donkey milk as well as processed milk samples (pasteurized and dried skimmed powder)	Species recognition based on amino acids, non-amino acids, and citric acid cycle metabolites	GC-MS	HCA	Euclidean distance and Ward’s method	[50]
Reconstituted milk vs. UHT milk	Different content of peptides, lipids, and nucleic acids	UPLC–Q-TOF-MS combined with UPLC–MS/MS	HCA	-	[51]
Cow milk	Fat globule characteristics (diameter, membrane surface, and yield), fat, protein, fatty acids, calcium content	IR (fat, protein, and lactose contents), GC (fatty acids composition), atomic absorption spectrophotometry (calcium content)	HCA	Euclidean distance	[52]
Cow, goat, buffalo, and camel milk	Different seasons of milk collection, based on sterols in milk fat of different species’ origin of milk	GC–MS-SIM	HCA	Euclidean distance	[53]

Abbreviations: CA = cluster analysis, CF-IRMS = continuous flow-isotope ratio mass spectrometer, CRDS = cavity ring-down spectroscopy, EA-IRMS = element analysis-isotope mass spectrometry, EIS = electrochemical impedance spectroscopy, FCM = fuzzy c-means, FTIR-ATR = Fourier transform infrared-attenuated total reflection, FTIR = Fourier transform infrared spectroscopy, GC = gas chromatography, GC-MS = gas chromatography-mass spectrometry, GC-MS-SIM = gas chromatography-mass spectrometry-single ion monitoring mode, HCA = hierarchical cluster analysis, ICP-MS = inductively coupled plasma mass spectrometry, ICP-OES = inductively coupled plasma emission spectroscopy, IR = infrared, IRMS = isotopic ratio mass spectrometry, UHT = ultra-high temperature, UPLC–MS/MS = UPLC–tandem mass spectrometry, UPLC–Q-TOF-MS = ultra-high performance liquid chromatography-quadrupole time-of-flight mass spectrometry.

**Table 2 foods-12-00139-t002:** Main classification methods cited.

Classification Method	Extended Name	Abbreviation
Discriminant	Partial least squares-discriminant analysis	PLS-DA
	Orthogonal partial least squares-discriminant analysis	OPLS-DA
	One class-partial least squares	OC-PLS
	Quadratic discriminant analysis	QDA
	Random forest	RF
	Support vector machine	SVM
	Linear discriminant analysis	LDA
	k-nearest neighbors	kNN
	Extreme learning machine	ELM
	Ensemble of extreme learning machine	EELM
Class-modeling	Soft independent modeling of class analogy	SIMCA
	Data-driven soft independent modeling of class analogy	DD-SIMCA
	Unequal class models	UNEQ

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
