# Peer review of "How Chemometrics Can Fight Milk Adulteration"

_foods, 2022, doi:10.3390/foods12010139_

Round 1

Reviewer 1 Report

This paper reviews the chemometrics-based approaches for milk adulteration detection. The clustering, classification, and regression were introduced to several analytical data. Moreover, the spectroscopy, chromatography, and electrochemical sensors for milk adulteration were described. In addition, a critical discussion was presented on how to face adulteration issues. This manuscript is interesting and provides some useful advice.

Some suggestions about this paper are as follows:

1. The abstract contains trivial statements. Such information can be easily found on Wikipedia. The content should be reduced in length and focus on current challenges and your work.

2. The detailed workflow of different techniques was preferred in the “Chemometric approaches” part, which will be easy for readers to understand. Paragraphs should be added to describe the workflow in part.

3. The clustering method mainly introduced in this manuscript was HCA. However, comparing and discussing HCA and other methods should be added. In addition, which method is suitable or unsuitable when facing different adulteration issues? This part of the content is critical for adulteration analysis.

4. The implications present in this manuscript were just listed without deep discussion. The commons and differences among these methods should be focused.

Author Response

This paper reviews the chemometrics-based approaches for milk adulteration detection. The clustering, classification, and regression were introduced to several analytical data. Moreover, the spectroscopy, chromatography, and electrochemical sensors for milk adulteration were described. In addition, a critical discussion was presented on how to face adulteration issues. This manuscript is interesting and provides some useful advice.

We would like to thank the reviewer for the feedback.

Some suggestions about this paper are as follows:

  1. The abstract contains trivial statements. Such information can be easily found on Wikipedia. The content should be reduced in length and focus on current challenges and your work.

The abstract has been edited according to the reviewer suggestions.

  1. The detailed workflow of different techniques was preferred in the “Chemometric approaches” part, which will be easy for readers to understand. Paragraphs should be added to describe the workflow in part.

Section 3 has been divided into paragraphs and subparagraphs to fully describe development and validation of chemometrics.

  1. The clustering method mainly introduced in this manuscript was HCA. However, comparing and discussing HCA and other methods should be added. In addition, which method is suitable or unsuitable when facing different adulteration issues? This part of the content is critical for adulteration analysis.

We have replied to this in the conclusions section at the final paragraph, so please see the rows 902-910.

  1. The implications present in this manuscript were just listed without deep discussion. The commons and differences among these methods should be focused.

We agree with the reviewer. We added some discussion regarding differences among the chemometric methods explored in this review (305-308, 358-373, and 524-532). We also enriched the conclusions with some additional comments (rows 902-910).

Reviewer 2 Report

Recommendation: minor revision for Foods.

 This paper summarizes the use of different stoichiometric techniques and spectroscopic, chromatographic, and electrochemical sensors to combat milk adulteration. The article is well summarized, but some topics need to be improved in the submitted paper:

1.     Some of the methods used for rapid and on-site detection can be supplemented.

2.     Authors should discuss the advantages and disadvantages of each approach and present their own views.

Author Response

This paper summarizes the use of different stoichiometric techniques and spectroscopic, chromatographic, and electrochemical sensors to combat milk adulteration. The article is well summarized, but some topics need to be improved in the submitted paper:

We would like to thank the reviewer for the valuable comments.

  1. Some of the methods used for rapid and on-site detection can be supplemented.

We added some comments about methods used for rapid and on-site detection in a new paragraph (rows 859-885).

  1. Authors should discuss the advantages and disadvantages of each approach and present their own views.

We agree with the reviewer. We added some discussion regarding differences among the chemometric methods explored in this review (305-308, 358-373, and 524-532).

We have also added a concluding paragraph in the conclusions section (rows 902-910)

Reviewer 3 Report

The review concerns an important topic in the use of NMR spectroscopy in the field of dairy product analysis applied to milk adulteration. It is written well, has a high level of scientific novelty and will be definitely interesting to the reader. I think that this paper can be accepted after correction of some minor issues. These are as follows:

NMR analysis of milk samples in connection with metabolomics is mentioned, however, only several papers are cited. At the same time there are much more works on this topic, for example, those of Brescia et al. 2004 (https://doi.org/10.1007/s11746-004-0918-3), Sundekilde et al 2013 (https://doi.org/10.3390/metabo3020204), Yanibada et al. 2018 (https://doi.org/10.1016/j.heliyon.2018.e00856) and especially Tomassini et al. 2019 (https://doi.org/10.1080/14786419.2018.1462183). Please consider including in the text, tables, and bibliography.

Author Response

The review concerns an important topic in the use of NMR spectroscopy in the field of dairy product analysis applied to milk adulteration. It is written well, has a high level of scientific novelty and will be definitely interesting to the reader. I think that this paper can be accepted after correction of some minor issues. These are as follows:

NMR analysis of milk samples in connection with metabolomics is mentioned, however, only several papers are cited. At the same time there are much more works on this topic, for example, those of Brescia et al. 2004 (https://doi.org/10.1007/s11746-004-0918-3), Sundekilde et al 2013 (https://doi.org/10.3390/metabo3020204), Yanibada et al. 2018 (https://doi.org/10.1016/j.heliyon.2018.e00856) and especially Tomassini et al. 2019 (https://doi.org/10.1080/14786419.2018.1462183). Please consider including in the text, tables, and bibliography.

We would like to thank the reviewer for the valuable feedback and suggestions. We added all the papers suggested by the reviewer in the text, tables, and bibliography except the one by Brescia et al. 2004, since our review focuses on papers written in the last years. We obviously added some comments related to these papers in rows 346-354.